# Gender disparity in the individual attitude toward longevity among Japanese population: Findings from a national survey

**Ruoyan Gai Tobe●\*, Nobuyuki Izumida**

Department of Social security Empirical Research, National Institute of Population and Social Security Research, Tokyo, Japan

\* gai-ruoyan@ipss.go.jp

## Abstract

The unprecedented population aging brings profound influences to the social values of longevity. The individual attitudes toward the expended life time deserves scrutiny, as it reflects the impacts of social networks and social welfare on people's life and wellbeing. This study aims to examine whether and how gender disparity is affecting the individual anticipation to longevity among Japanese citizen. We used the dataset of National Survey on Social Security and Peoples Life implemented in 2017 to calculate the odds ratios (OR) of the individual anticipation to longevity. Besides gender, other demographic characteristics, physical and mental health, the experience of nursing care for the elderly, financial conditions and social networks are examined by performing the multilevel mixed-effects logistic regression analysis. The results indicate the robust effects of gender disparity on the individual aspiration for longevity. The proportion of those who inclined the positive statement on longevity was estimated to be 69.7% (95% CI: 68.6% - 70.9%) in the whole population, and 70.9% (95% CI: 69.4% - 72.5%) and 68.7% (95% CI: 67.1% - 70.2%) in male and female, respectively. Besides gender, independent factors significantly affecting the individual valuation of longevity include age, annual household income, the experience of nursing care, household saving, having a conversation with others and the availability of reliable partner(s) for relevant supports; while the common factors affecting the outcome variable were self-perceived health status and mental distress measured by K6. The interaction of gender and these significant factors were determined as well. In conclusion, with relevant representativeness and quality of data source, this analysis adds knowledge on gender disparity in the individual anticipation on longevity. The findings are suggestive to reform the social security system in the super aged society.

## Introduction

Longevity is the achievement of public health and social development. In Japan, one of the most super-aged society in the world, the percentage of the elderly (aged > = 65 years) in the overall population has reached 28% in 2018 and is expected to exceed 38% in 2065. The

**Data Availability Statement:** The confidential data are held in National Institute of Population and Social Security Research, Japan (URL: http://www.ipss.go.jp/site-ad/index_english/Survey-e.asp). The full dataset will be only be available after approval

for researchers who meet the criteria for the access. For inquiry or request of the survey data, please contact the link below and mention "application to the secondary data-use of The National Survey on Social Security and People's Life 2017". http://www.ipss.go.jp/mail/e_sendmail/mail.html.

**Funding:** The author(s) received no specific funding for this work.

**Competing interests:** The authors have declared that no competing interests exist.

average life expectancy continues to increase during the past decades and has reached 81.3 years and 87.3 years for men and women in 2018, respectively [1]. The fertility remains low, and the ratio of the elderly to the population at working age is increasing. Meanwhile, the average family size is projected to decrease from 2.33 persons per household to 2.08 persons, and the proportion of single households among those headed by those 65 years and older will increase from 36.0% to 44.2% between 2015 and 2040 [2]. Such the unprecedented changes in demographics and family structure brings profound influences to society in multiple ways. Labor supply is reducing and productivity is slowing down. The soaring expenditures for health care and long-term care is challenging the sustainability of social welfare system. Women's family role is changing with improved employment status, delayed age at marriage and decreased fertility.

In a super-aged society, is longevity still a blessing, what influence the social value regarding the length of lifespan, and how should the social security system respond to people's desires on their life? So far, seldom have empirical studies been conducted to figure out these essential questions related to population health and wellbeing. As an emerging issue, the individual attitude on the expended lifetime deserves scrutiny, as it reflects life satisfaction and wellbeing of people and the impact of social networks and social welfare on that. Tackling population aging worldwide, the World Health Organization (WHO) has emphasized actions to promote healthy and active aging, such as implementing proactive interventions to maintain physical and mental capabilities, fostering productivity and social engagement of the elderly people and mobilizing all relevant sectors to create an age-friendly society that encompasses health system, long-term care system, pension systems and social environment [3, 4]. Healthy and active aging entails delaying the onset of diseases and disabilities and compressing morbidities, as physical and mental functioning plays a principle role on the individual wellbeing and productivity [5, 6]. Preventive measures of typical age-related conditions that contribute to loss of independence among the elderly people, such as frailty / pre-frailty, sequelae of lifestyle-related diseases and dementia, have been implemented with the proved effectiveness in prolonging healthy life expectancy [7–12]. In the super-aged society, living healthier signifies not only the length but also the quality of life, and brings benefits to the sustainability of the social security system.

Whereas it remains throughout the entire life course in terms of various social and economic determinants affecting health, functioning and wellbeing, gender disparity cannot be neglected in late life and progression of disability in particular [13–15]. As women have much longer life expectancy than men, women at their old age are more likely to be widowed compared to their male counterparts. Those living alone are exposed to higher risks of physical and mental health consequences and lack of social and financial supports [16, 17]. Previous studies indicated epidemiological differences between men and women in Japan and other settings of the world, such as longer years both with and without disability, higher incidence of frailty and pre-frailty while lower mortality at the same frailty level, and slower progress of disability and longer duration in a disabled state in women [18–21]. Underlying this gender disparity are the complex interaction of biological and socioeconomic factors, which are accumulated along the life course and manifested in the disadvantages of women [22, 23].

With these concerns, this study aimed to explore the individual anticipation to longevity and the affecting factors in Japan by analyzing the latest large-scale national survey. In our hypothesis, the effects of gender disparity were supposed to be a key explanatory factor based on its impact on population aging and health. From this fundamental research question, we aim to explore issues associating with the longevity desire at both public and private / individual level. It is worth determining that how the social network and the social security system

influencing life and wellbeing of population at all ages and how gender disparity appears in these matters.

## Methods

### Study population

The data derived from the latest National Survey on Social Security and People's Life in July 2017, which is a national survey implemented in every 5 years, aiming to investigate people's lives, family relations, and socioeconomic activities of individuals and to scrutinize the roles that social security benefits and social networks play. The full dataset merged the valid response from 10,369 households and 19,800 adult individuals (aged 18 years or older) living in 300 municipalities, which were officially selected at random from the study settings of Comprehensive Survey of Living Conditions in 2017, another national survey implemented by the Ministry of Health, Labor and Welfare, Japan. The study settings covered all 47 prefectures of the country. The survey included the household questionnaire and the individual questionnaire (that surveyed each member of the household), and both of these questionnaires were distributed by the trained survey staff, self-completed and sealed in the envelopes by the respondents and finally collected by the staff. The valid response rate was 63.5% and 75.0% for the household and the individual questionnaire, respectively. **Table 1** summarized major demographic characteristics of the respondents. Details of the survey were described in the official report [24].

### Variables and covariates

The outcome variable, the individual anticipation for longevity was surveyed in the individual questionnaire, with four scaled options "1. Strongly agree, 2. Somewhat agree, 3. Somewhat disagree, 4. Strongly disagree" to the question "Living long is a good thing". The answers were then dichotomized into strongly agree / somewhat agree versus strongly disagree / somewhat disagree in multilevel analysis.

The explanatory variable, gender disparity as originated from distinctions in biology, psychology and sociocultural practices, referred to the different distribution of the answers in the male and female. As for potential confounding variables, we introduced a series of covariates in the multivariate models, including age (which is continuous variable in years and then categorized into age groups), whether or not having a spouse, ownership of house, whether or not having children, employment status, annual household income (which is continuous variable in Japanese Yen and then categorized into the low, middle and high income groups), subjective health status (five scaled options: 1. Excellent, 2. Good, 3. Fair, 4. Poor, 5. Bad), physical limitation due to poor health (three scaled options: 1. Very limited, 2. Limited, 3. Not limited), mental distress measured by self-administrated Kessler Psychological Distress Scale (K6) (mental distress is scored and categorized into three levels: 1. No or mild mental distress with K6 score<5, 2. Moderate mental distress with $5< =$ K6 score<13, 3. Serious mental distress with K6 score$> = 13$) [25], the current or / and previous experience of nursing care for the elderly, whether or not having household savings, whether or not having household debt, self-perceived financial status (five scaled options: 1. Very well off, 2. Well off, 3. Fair, 4. Badly off, 5. Very badly off), whether or not having a conversation with others, and availability of reliable partner(s) for nursing care, for consulting on crucial events, for listening to complaints, for sharing joys and sorrows of life, for financial supports and for helps in daily life (**Table 2**). Missing data, which were recorded as "9" in the original dataset, were shifted to "." for data analysis.

**Table 1. Basic demographic characteristics of participants (1).**

| | | n | % |
|---|---|---|---|
| **Type of the household** | Male one-person household | 1,079 | 5.45 |
| | Female one-person household | 1,233 | 6.23 |
| | Household consisting of a couple | 4,539 | 22.92 |
| | Household consisting of a couple and dependent offspring | 3,889 | 19.64 |
| | Household consisting of a single parent and dependent offspring | 284 | 1.43 |
| | Household consisting of three generations | 1,088 | 5.49 |
| | Household whose structure is unknown | 334 | 1.69 |
| | Other type of household | 7,354 | 37.14 |
| **Age** | 20 years and younger | 377 | 1.9 |
| | 20–29 years | 1,752 | 8.85 |
| | 30–39 years | 2,600 | 13.13 |
| | 40–49 years | 3,375 | 17.05 |
| | 50–59 years | 3,127 | 15.79 |
| | 60–69 years | 4,051 | 20.46 |
| | 70–79 years | 2,939 | 14.84 |
| | 80–89 years | 1,380 | 6.97 |
| | 90 years and older | 199 | 1.01 |
| **Gender** | Male | 9,446 | 47.71 |
| | Female | 10,354 | 52.29 |
| **Spouse** | Without spouse | 6,736 | 34.71 |
| | With spouse | 12,669 | 65.29 |
| **Offspring** | Having offspring | 13,629 | 72.21 |
| | Not having offspring | 5,245 | 27.79 |
| **Educational background** | Primary and middle school | 2,395 | 12.46 |
| | High school | 7,931 | 41.26 |
| | Junior college | 2,016 | 10.49 |
| | University and graduate school | 4,819 | 25.07 |
| | Others | 2,062 | 10.73 |
| **Annual household income** | Low | 6,132 | 33.42 |
| | Middle | 6,120 | 33.36 |
| | High | 6,094 | 33.22 |

## Statistical analysis

First, we assessed the distribution of the individual anticipation for longevity in each variable and calculated crude odds ratios. Then, we performed multilevel mixed-effects logistic regression to explore the relationship between the individual anticipation for longevity and gender, adjusting for covariates in terms of demographic characteristics, physical and mental health, the experience of nursing care for the elderly, financial conditions and social networks (level 1: 19,800 individuals, level 2: 300 municipalities; level 3: 47 prefectures). Intra-class cluster correlation was controlled (municipality and prefecture level). By these models, we acquired adjusted odds ratios and the proportion of agreement / disagreement to the statement with a 95% confidence interval (95% CI). We used Stata 15.1 for all analysis.

## Results

We analyzed 9,446 men and 10,354 women aged above 18 years living in 300 municipalities of Japan. In a total of 19,800 adult participants, those inclining with the statement "longevity is a good thing" accounted for 68.8%.

Table 3 demonstrated the crude and adjusted ORs of the positive response regarding the individual anticipation for longevity, which derived from the univariate analysis and the multi-level regression model, respectively. The multilevel regression model in the whole population predicted the proportion of agreement to the positive statement on longevity to be 68.9% (95% CI: 68.2% - 69.5%). Age, gender, annual household income, status of qualifying for the public assistance, self-perceived health status, mental distress, the current or /and previous experience of nursing care for the elderly, whether or not having a conversation with others, and the availability of reliable partner(s) for nursing care, for sharing joys and sorrows of life, for financial supports and for helps in daily life significantly affected the outcome variable.

In the stratified model by gender, the proportion of the positive response is 70.9% (95% CI: 69.4% - 72.5%) and 68.7% (95% CI: 67.1% - 70.2%) in male and female, respectively. In male interviewees, those with annual household income at the middle level and those with worse physical and mental health status tended to have higher proportion in disagreement to the positive statement on longevity, while those having household savings, those having reliable partner(s) for sharing the joys and sorrows of life and for helps in daily life tended to have a positive attitude to longevity. On the other hand, in female interviewees, among those tended to not incline the positive statement on longevity were those aged between 40 to 70 years, those with worse physical and mental health status, and those having the current or / and previous experience of nursing care for the elderly, while those having a conversation with others and those having reliable partner(s) for nursing care were more likely to incline the positive statement. Table 4 summarized the results of the multilevel regression model by gender strata. There is a difference in variables significantly affecting the outcome variable by gender, including age, annual household income, the experience of nursing care, household saving, status of qualifying for the public assistance, having a conversation with others and the availability of reliable partner(s) for relevant supports; while the common factors affecting the outcome variable were self-perceived health status and mental distress measured by K6. The proportion of agreement to the positive statement was gradually declining from the optimal to the worst physical and mental health status.

We also explored the interactions of gender and those significant variables in the multilevel regression model, as summarized in Table 5. As the results, female aged above 40 years, female with the experience of nursing care, female not having a conversation with others, female not having reliable partner(s) for nursing care and for financial support were less likely to report the positive response compared to others; on the other hand, male having household savings and male having reliable partner(s) for helps in daily life were more likely to do so. In terms of annual household income, compared to male with the low-level income, male with the middle-level income and female at different income levels were less likely to have the positive response. Regarding self-perceived health status, the significant difference of the OR compared to male reporting excellent health status appeared in both male and female who reported fair or even worse health status. As for K6, compared to male reporting no or mild mental distress, male reporting moderate to serious mental distress and female at all these mental distress levels were less likely to respond positively.

## Discussion

To our knowledge, this is the first empirical study to investigate the individual anticipation to longevity in the population of a super-aging society, with the percentage of the positive response in the overall population as well as the relevant subgroups reported. With a wide penetration of population aging in the public, concerns on health and wellbeing of the late life have raised in not only the elderly, but also the overall population in Japan. The individual

**Table 2. Basic demographic characteristics of participants (2).**

|  |  | n | % |
|---|---|---|---|
| **Subjective health status** | Excellent | 4,620 | 23.43 |
|  | Good | 4,404 | 22.33 |
|  | Fair | 7,317 | 37.11 |
|  | Poor | 2,799 | 14.2 |
|  | Bad | 578 | 2.93 |
| **physical limitation due to poor health** | Very limited | 976 | 5 |
|  | Limited | 3,536 | 18.12 |
|  | Not limited | 15,006 | 76.88 |
| **Kessler Psychological Distress Scale (K6)** | No or mild mental distress (K6 score<5) | 9,907 | 51.45 |
|  | Moderate mental distress (5< = K6 score<13) | 5,787 | 30.05 |
|  | Serious mental distress (K6 score> = 13) | 3,561 | 18.49 |
| **Experience of nursing care for the elderly** | No | 7,640 | 65.36 |
|  | Yes | 4,050 | 34.64 |
| **Qualifying for the public assistance** | No | 18,843 | 98.76 |
|  | Yes | 237 | 1.24 |
| **Household savings** | No | 4,456 | 25.04 |
|  | Yes | 13,339 | 74.96 |
| **Household debts** | No | 12,387 | 67.81 |
|  | Yes | 5,879 | 32.19 |
| **Self-perceived financial status** | Very well off | 383 | 1.97 |
|  | Well off | 1,763 | 9.09 |
|  | Fair | 10,654 | 54.93 |
|  | Badly off | 4,954 | 25.54 |
|  | Very badly off | 1,642 | 8.47 |
| **conversation with others** | No | 1,700 | 8.79 |
|  | Yes | 17,647 | 91.21 |
| **Availability of reliable partner(s) for nursing care** | No | 6,275 | 35.87 |
|  | Yes | 11,218 | 64.13 |
| **Availability of reliable partner(s) for consulting on crucial events** | No | 2,115 | 11.54 |
|  | Yes | 16,206 | 88.46 |
| **Availability of reliable partner(s) for listening to complaints** | No | 2,194 | 11.92 |
|  | Yes | 16,205 | 88.08 |
| **Availability of reliable partner(s) for sharing joys and sorrows of life** | No | 1,545 | 8.41 |
|  | Yes | 16,819 | 91.59 |
| **Availability of reliable partner(s) for financial supports** | No | 7,373 | 40.11 |
|  | Yes | 11,007 | 59.89 |
| **Availability of reliable partner(s) for helps in daily life** | No | 2,704 | 14.76 |
|  | Yes | 15,613 | 85.24 |
| **The individual anticipation for longevity** | Strongly agree | 4,816 | 24.65 |
|  | Somewhat agree | 8,634 | 44.19 |
|  | Somewhat disagree | 5,287 | 27.06 |
|  | Strongly disagree | 801 | 4.1 |

**Table 3. Crude and adjusted OR of the individual anticipation for longevity.**

| longevity | | OR (crude) | 95% CI | | p | OR (adjusted) | 95% CI | | p |
|---|---|---|---|---|---|---|---|---|---|
| **Age** | <40 years | ref. | | | | ref. | | | |
| | 40–70 years | 0.666 | 0.615 | 0.720 | **0.000** | 0.777 | 0.667 | 0.905 | **0.001** |
| | >70 years | 0.570 | 0.521 | 0.624 | **0.000** | 0.791 | 0.645 | 0.970 | **0.024** |
| **Sex** | Male | ref. | | | | ref. | | | |
| | Female | 0.894 | 0.842 | 0.950 | **0.000** | 0.849 | 0.763 | 0.946 | **0.003** |
| **Spouse** | Without spouse | ref. | | | | ref. | | | |
| | With spouse | 1.241 | 1.164 | 1.322 | **0.000** | 1.027 | 0.895 | 1.179 | 0.705 |
| **Employment status** | Currently employed | ref. | | | | ref. | | | |
| | Seeking work | 0.735 | 0.651 | 0.830 | **0.000** | 1.132 | 0.915 | 1.399 | 0.254 |
| | Not working | 0.822 | 0.768 | 0.881 | **0.000** | 1.054 | 0.924 | 1.203 | 0.431 |
| **Annual household income** | Low | ref. | | | | ref. | | | |
| | Middle | 1.110 | 1.029 | 1.198 | **0.007** | 0.860 | 0.756 | 0.980 | **0.023** |
| | High | 1.431 | 1.324 | 1.547 | **0.000** | 1.040 | 0.902 | 1.199 | 0.590 |
| **Ownership of house** | Owned | ref. | | | | ref. | | | |
| | Rented or other | 0.932 | 0.865 | 1.003 | 0.06 | 1.036 | 0.900 | 1.193 | 0.625 |
| **Offspring** | Having offspring | ref. | | | | ref. | | | |
| | Not having offspring | 0.968 | 0.903 | 1.037 | 0.356 | 0.882 | 0.758 | 1.026 | 0.103 |
| **Qualifying for the public assistance** | Yes | ref. | | | | ref. | | | |
| | No | 2.106 | 1.626 | 2.728 | **0.000** | 0.502 | 0.281 | 0.895 | **0.020** |
| **Subjective health status** | Excellent | ref. | | | | ref. | | | |
| | Good | 0.864 | 0.780 | 0.957 | **0.005** | 1.017 | 0.864 | 1.198 | 0.837 |
| | Fair | 0.454 | 0.416 | 0.495 | **0.000** | ref. | 0.514 | 0.691 | **0.000** |
| | Poor | 0.259 | 0.233 | 0.287 | **0.000** | 0.417 | 0.340 | 0.511 | **0.000** |
| | Bad | 0.163 | 0.135 | 0.196 | **0.000** | 0.367 | 0.252 | 0.535 | **0.000** |
| **physical limitation due to poor health** | Very limited | ref. | | | | ref. | | | |
| | Limited | 1.555 | 1.347 | 1.796 | **0.000** | 1.093 | 0.833 | 1.433 | 0.521 |
| | Not limited | 2.602 | 2.281 | 2.968 | **0.000** | 0.965 | 0.734 | 1.268 | 0.797 |
| **Kessler Psychological Distress Scale (K6)** | No or mild mental distress | ref. | | | | ref. | | | |
| | Moderate mental distress | 0.671 | 0.624 | 0.722 | **0.000** | 0.778 | 0.692 | 0.875 | **0.000** |
| | Serious mental distress | 0.307 | 0.283 | 0.333 | **0.000** | 0.441 | 0.382 | 0.510 | **0.000** |
| **Experience of nursing care for the elderly** | No | ref. | | | | ref. | | | |
| | Yes | 0.675 | 0.622 | 0.732 | **0.000** | 0.748 | 0.669 | 0.836 | **0.000** |
| **Household savings** | No | ref. | | | | ref. | | | |
| | Yes | 1.388 | 1.292 | 1.492 | **0.000** | 1.107 | 0.968 | 1.265 | 0.138 |
| **Household debts** | No | ref. | | | | ref. | | | |
| | Yes | 1.188 | 1.109 | 1.272 | **0.000** | 1.102 | 0.977 | 1.244 | 0.114 |
| **Self-perceived financial status** | Very well off | ref. | | | | ref. | | | |
| | Well off | 1.384 | 1.075 | 1.782 | **0.012** | 1.182 | 0.802 | 1.743 | 0.399 |
| | Fair | 0.999 | 0.794 | 1.256 | **0.991** | 0.998 | 0.698 | 1.426 | 0.989 |
| | Badly off | 0.656 | 0.520 | 0.828 | **0.000** | 0.854 | 0.591 | 1.235 | 0.402 |
| | Very badly off | 0.381 | 0.298 | 0.487 | **0.000** | 0.705 | 0.473 | 1.051 | 0.086 |
| **Conversation with others** | No | ref. | | | | ref. | | | |
| | Yes | 2.040 | 1.843 | 2.257 | **0.000** | 1.225 | 1.004 | 1.494 | **0.045** |
| **Availability of reliable partner(s) for nursing care** | No | ref. | | | | ref. | | | |
| | Yes | 1.534 | 1.436 | 1.639 | **0.000** | 1.224 | 1.089 | 1.376 | **0.001** |

*(Continued)*

**Table 3.** (Continued)

| longevity | | OR (crude) | 95% CI | | p | OR (adjusted) | 95% CI | | p |
|---|---|---|---|---|---|---|---|---|---|
| Availability of reliable partner(s) for consulting on crucial events | No | ref. | | | | ref. | | | |
| | Yes | 2.351 | 2.143 | 2.578 | **0.000** | 1.177 | 0.963 | 1.437 | 0.111 |
| Availability of reliable partner(s) for listening to complaints | No | ref. | | | | ref. | | | |
| | Yes | 2.084 | 1.902 | 2.283 | **0.000** | 1.013 | 0.821 | 1.249 | 0.906 |
| Availability of reliable partner(s) for sharing joys and sorrows of life | No | ref. | | | | ref. | | | |
| | Yes | 2.758 | 2.480 | 3.066 | **0.000** | 1.361 | 1.064 | 1.739 | **0.014** |
| Availability of reliable partner(s) for financial supports | No | ref. | | | | ref. | | | |
| | Yes | 1.653 | 1.551 | 1.761 | **0.000** | 1.134 | 1.015 | 1.267 | **0.026** |
| Availability of reliable partner(s) for helps in daily life | No | ref. | | | | ref. | | | |
| | Yes | 2.007 | 1.845 | 2.183 | **0.000** | 1.291 | 1.093 | 1.525 | **0.003** |

will-to-live reflected the well-being of older people and closely associated with health and well-being of the elderly [26–29]. Compared to previous studies on desires and will-to-live focusing on the older people [26–33], our study had a good sampling frame to achieve the representativeness of the overall population in the country, and by applying plenty of questions that were designed to scrutinize various aspects of people's life and social security issues, we were able to examine the effects of diverse covariates in details.

We identified that the attitude to a longer lifetime was diversified by gender, changing with age and affected by physical and mental health status and available social supports and financial resources. It was consistent to findings of a narrative review of previous studies implemented in different settings to assess longevity preference and motivation, suggesting among the determinants for longevity motivation are 1) contextual influences such as culture, age, gender, personal experiences, 2) health functioning and 3) personal beliefs such as attitudes toward aging, religiosity, future perspectives and death acceptance [30].

In general, men have higher proportion of the positive statement on longevity compared to their female counterparts. Affecting factors were different in male and female groups as well. The result underlined the impact of gender disparity on the individual desires to longevity, corresponding to that on population health and aging as the results of the combination between biological characteristics and social factors related to health behaviors, social role, lifestyle, and life experiences [23]. The revealed gender difference of the attitude on longevity could be explained by the "health-survival paradox", which refers to the fact that the advantage in life expectancy, prevalence of chronic conditions and mortality among women tends to be offset by the disadvantage in the duration of disability especially in settings with high life expectancy [34]. In Japan, the duration of time spent with disability in women was longer in men [35]. The number of years living with disability is increasing, with the (life expectancy–healthy life expectancy) / life expectancy figure from 10.5% in 1990 to 12.5% in 2005 for women, affecting older women's quality of life [36]. The health-survival paradox, in particular the longer duration living with disability, suggested women may have more needs to both formal and informal care. Concerns on the late life, as well as the need of life plan for the life span at individual / private and public level may raise from these demographic and epidemiological characteristics and challenges [37].

Health is the most fundamental element of wellbeing and paramount to the planning for the life span at both individual and society level, which was demonstrated by our results on the robust impact of physical and mental health status on the individual desires on longevity.

**Table 4. Gender disparity.**

| | | Male | | | | Female | | | |
|---|---|---|---|---|---|---|---|---|---|
| **Longevity** | | OR | 95% CI | | p | OR | 95% CI | | p |
| **Age** | <40 years | ref. | | | | ref. | | | |
| | 40–70 years | 0.838 | 0.676 | 1.039 | 0.107 | 0.725 | 0.584 | 0.901 | **0.004** |
| | >70 years | 0.767 | 0.569 | 1.036 | 0.084 | 0.777 | 0.584 | 1.034 | 0.083 |
| **Spouse** | Without spouse | ref. | | | | ref. | | | |
| | With spouse | 1.145 | 0.905 | 1.449 | 0.260 | 0.954 | 0.800 | 1.137 | 0.597 |
| **Employment status** | Currently employed | ref. | | | | ref. | | | |
| | Seeking work | 1.051 | 0.754 | 1.465 | 0.770 | 1.203 | 0.907 | 1.596 | 0.199 |
| | Not working | 1.159 | 0.935 | 1.435 | 0.178 | 1.024 | 0.864 | 1.214 | 0.781 |
| **Annual household income** | Low | ref. | | | | ref. | | | |
| | Middle | 0.691 | 0.570 | 0.839 | **0.000** | 1.039 | 0.871 | 1.240 | 0.671 |
| | High | 0.865 | 0.699 | 1.069 | 0.179 | 1.194 | 0.985 | 1.448 | 0.072 |
| **Ownership of house** | Owned | ref. | | | | ref. | | | |
| | Rented or other | 1.008 | 0.829 | 1.225 | 0.939 | 1.070 | 0.880 | 1.300 | 0.498 |
| **Offspring** | Having offspring | ref. | | | | ref. | | | |
| | Not having offspring | 0.951 | 0.752 | 1.203 | 0.676 | 0.830 | 0.677 | 1.018 | 0.074 |
| **Qualifying for the public assistance** | Yes | ref. | | | | ref. | | | |
| | No | 0.646 | 0.314 | 1.329 | 0.235 | 0.349 | 0.131 | 0.930 | 0.035 |
| **Subjective health status** | Excellent | ref. | | | | ref. | | | |
| | Good | 1.052 | 0.831 | 1.331 | 0.674 | 0.996 | 0.792 | 1.253 | 0.975 |
| | Fair | 0.580 | 0.470 | 0.715 | **0.000** | 0.613 | 0.497 | 0.755 | **0.000** |
| | Poor | 0.435 | 0.326 | 0.581 | **0.000** | 0.398 | 0.298 | 0.530 | **0.000** |
| | Bad | 0.312 | 0.186 | 0.523 | **0.000** | 0.424 | 0.245 | 0.735 | **0.002** |
| **physical limitation due to poor health** | Very limited | ref. | | | | ref. | | | |
| | Limited | 1.189 | 0.806 | 1.756 | 0.383 | 1.060 | 0.723 | 1.552 | 0.767 |
| | Not limited | 1.124 | 0.761 | 1.660 | 0.557 | 0.883 | 0.599 | 1.303 | 0.532 |
| **Kessler Psychological Distress Scale (K6)** | No or mild mental distress | ref. | | | | ref. | | | |
| | Moderate mental distress | 0.793 | 0.670 | 0.939 | **0.007** | 0.777 | 0.660 | 0.915 | **0.002** |
| | Serious mental distress | 0.430 | 0.348 | 0.530 | **0.000** | 0.444 | 0.364 | 0.542 | **0.000** |
| **Experience of nursing care for the elderly** | No | ref. | | | | ref. | | | |
| | Yes | 0.882 | 0.744 | 1.046 | 0.149 | 0.661 | 0.569 | 0.768 | **0.000** |
| **Household savings** | No | ref. | | | | ref. | | | |
| | Yes | 1.214 | 1.006 | 1.466 | **0.044** | 1.022 | 0.844 | 1.237 | 0.825 |
| **Household debts** | No | | | | | ref. | | | |
| | Yes | 1.082 | 0.910 | 1.287 | 0.373 | 1.143 | 0.967 | 1.353 | 0.118 |
| **Self-perceived financial status** | Very well off | ref. | | | | ref. | | | |
| | Well off | 1.340 | 0.750 | 2.392 | 0.323 | 1.029 | 0.607 | 1.742 | 0.917 |
| | Fair | 1.041 | 0.612 | 1.770 | 0.882 | 0.933 | 0.573 | 1.518 | 0.780 |
| | Badly off | 0.854 | 0.494 | 1.477 | 0.573 | 0.833 | 0.504 | 1.378 | 0.477 |
| | Very badly off | 0.708 | 0.394 | 1.271 | 0.247 | 0.683 | 0.393 | 1.187 | 0.176 |
| **Conversation with others** | No | ref. | | | | ref. | | | |
| | Yes | 0.987 | 0.753 | 1.294 | 0.926 | 1.583 | 1.174 | 2.134 | **0.003** |
| **Availability of reliable partner(s) for nursing care** | No | ref. | | | | ref. | | | |
| | Yes | 1.138 | 0.959 | 1.350 | 0.139 | 1.299 | 1.104 | 1.527 | **0.002** |
| **Availability of reliable partner(s) for consulting on crucial events** | No | ref. | | | | ref. | | | |
| | Yes | 1.137 | 0.879 | 1.471 | 0.327 | 1.220 | 0.882 | 1.688 | 0.230 |
| **Availability of reliable partner(s) for listening to complaints** | No | ref. | | | | ref. | | | |

(*Continued*)

**Table 4.** (Continued)

| Longevity | | Male | | | | Female | | | |
|---|---|---|---|---|---|---|---|---|---|
| | | OR | 95% CI | | p | OR | 95% CI | | p |
| | Yes | 1.013 | 0.787 | 1.304 | 0.919 | 1.123 | 0.766 | 1.646 | 0.552 |
| Availability of reliable partner(s) for sharing joys and sorrows of life | No | ref. | | | | ref. | | | |
| | Yes | 1.357 | 1.006 | 1.831 | **0.046** | 1.369 | 0.881 | 2.128 | 0.163 |
| Availability of reliable partner(s) for financial supports | No | ref. | | | | ref. | | | |
| | Yes | 1.099 | 0.935 | 1.293 | 0.254 | 1.159 | 0.995 | 1.351 | 0.059 |
| Availability of reliable partner(s) for helps in daily life | No | ref. | | | | ref. | | | |
| | Yes | 1.353 | 1.080 | 1.696 | **0.009** | 1.253 | 0.977 | 1.605 | 0.075 |

There was generally no difference between the male and the female group on this finding. During the past decades, health promotion programs have been implemented at population level to extend healthy active aging in the country, emphasizing life-course approaches and the integrated care covering pre-clinical, clinical and post-clinical stages at the community level [11, 12, 38]. These measures are contributable to the increased healthy life expectancy and are underpinning the vitality of the super-aged society. On the other hand, it is worth noting that the gap between the overall life expectancy and healthy life expectancy, in other words, the duration living with disabilities, did not reduce [36], indicating that the well-functioned social security system is indispensable to provide the safety net for those living with disabilities and their caregivers.

The results reflected the concerns underlying the negative attitude on longevity, which were reinforced by personal experiences, financial status and social relationships as well. Those having the experience of caregiving for the elderly currently or previously tended to less likely have the positive response. When stratifying by gender, women with the caregiving experience, but not other counterparts, had the significant low OR. Traditionally, women are the key player of the nursing care for the elderly because of the social norm, providing primarily home-based care and supporting facility-based care [39]. Because of the longer duration living with disabilities as discussed above, women are also the major recipient of the nursing care. The experience of caregiving may bring tremendous long-lasting influences on physical and mental health, quality of life, productivity / income and social life [40]. According to a theory of Feminist Gerontology, caregivers for the old, sick and disabled may experience loss of power, status, and respect, resulting in financial and subsequent health disparities across the life course and into old age, and the inequities experienced by women as both caregivers and recipient of care contributed to health, functioning and wellbeing of the elderly, especially older women [40–42]. The results of our study added the relevant empirical evidence on gender and aging in especially the sociocultural and policy context of Japan.

Our results that female not having reliable partner(s) for nursing care were less likely to have the positive attitude on a longer lifetime suggested the substantial concerns about family caregiving in late life especially among female. In Japan, although the nursing care has been institutionalized and regarded as a public service rather than a private duty since the establishment of the Long-Term Care Insurance System in 2000, family caregivers still play the essential role. The percentage of households in which family members are the major caregiver has not been changed significantly [43]. Regarding family caregiving, older women are more likely to rely on non-spousal caregivers [44–46]. On the other hand, the dramatic change of demographical and family structure resulted in shrinking supply of family caregivers [47]. The trajectory of essential family functions in caregiving of the elderly people is being weakened. Moreover, the increasing costs of facility-based care and the promotion of home-based and

**Table 5. Interactions of major factors and gender.**

| Age | <40 years # male | ref. | | | |
|---|---|---|---|---|---|
| | <40 years # female | 1.035 | 0.827 | 1.296 | 0.765 |
| | 40–70 years # male | 0.864 | 0.709 | 1.053 | 0.148 |
| | *40–70 years # female* | *0.695* | *0.569* | *0.850* | *0.000* |
| | >70 years # male | 0.868 | 0.676 | 1.114 | 0.267 |
| | *>70 years # female* | *0.751* | *0.587* | *0.961* | *0.023* |
| Offspring | having offspring # male | ref. | | | |
| | having offspring # female | 0.880 | 0.729 | 1.064 | 0.187 |
| | *not having offspring # male* | *0.862* | *0.764* | *0.972* | *0.015* |
| | *not having offspring # female* | *0.752* | *0.619* | *0.914* | *0.004* |
| Annual household income | low # male | ref. | | | |
| | *middle # male* | *0.697* | *0.579* | *0.838* | *0.000* |
| | high # male | 0.900 | 0.739 | 1.094 | 0.290 |
| | *low # female* | *0.681* | *0.565* | *0.822* | *0.000* |
| | *middle # female* | *0.704* | *0.583* | *0.850* | *0.000* |
| | *high # female* | *0.790* | *0.647* | *0.963* | *0.020* |
| Household savings | no # male | ref. | | | |
| | *yes # male* | *1.242* | *1.043* | *1.479* | *0.015* |
| | no # female | 1.003 | 0.814 | 1.236 | 0.977 |
| | yes # female | 1.021 | 0.856 | 1.218 | 0.820 |
| Experience of nursing care for the elderly | no # male | ref. | | | |
| | yes # male | 0.913 | 0.775 | 1.076 | 0.277 |
| | no # female | 0.965 | 0.850 | 1.096 | 0.583 |
| | *yes # female* | *0.632* | *0.551* | *0.725* | *0.000* |
| conversation with others | no # male | ref. | | | |
| | yes # male | 0.972 | 0.757 | 1.248 | 0.824 |
| | *no # female* | *0.541* | *0.382* | *0.768* | *0.001* |
| | yes # female | 0.868 | 0.674 | 1.119 | 0.275 |
| Availability of reliable partner(s) for nursing care | no # male | ref. | | | |
| | yes # male | 1.137 | 0.968 | 1.335 | 0.118 |
| | *no # female* | *0.794* | *0.668* | *0.944* | *0.009* |
| | yes # female | 1.017 | 0.864 | 1.198 | 0.838 |
| Availability of reliable partner(s) for sharing joys and sorrows of life | no # male | ref. | | | |
| | yes # male | 1.300 | 0.997 | 1.695 | 0.052 |
| | no # female | 0.716 | 0.491 | 1.044 | 0.082 |
| | yes # female | 1.134 | 0.864 | 1.488 | 0.366 |
| Availability of reliable partner(s) for financial supports | no # male | ref. | | | |
| | yes # male | 1.052 | 0.903 | 1.226 | 0.514 |
| | *no # female* | *0.796* | *0.680* | *0.930* | *0.004* |
| | yes # female | 0.956 | 0.822 | 1.113 | 0.563 |
| Availability of reliable partner(s) for helps in daily life | no # male | ref. | | | |
| | *yes # male* | *1.236* | *1.006* | *1.518* | *0.044* |
| | no # female | 0.805 | 0.621 | 1.045 | 0.103 |
| | yes # female | 1.074 | 0.871 | 1.325 | 0.502 |
| Subjective health status | excellent # male | ref. | | | |
| | good # male | 1.083 | 0.859 | 1.366 | 0.501 |
| | *fair # male* | *0.588* | *0.481* | *0.720* | *0.000* |
| | *poor # male* | *0.439* | *0.338* | *0.572* | *0.000* |

*(Continued)*

**Table 5.** (Continued)

| | | | | | |
|---|---|---|---|---|---|
| | *bad # male* | *0.280* | *0.173* | *0.453* | *0.000* |
| | excellent # female | 0.885 | 0.701 | 1.117 | 0.304 |
| | good # female | 0.854 | 0.680 | 1.071 | 0.172 |
| | *fair # female* | *0.520* | *0.425* | *0.638* | *0.000* |
| | *poor # female* | *0.353* | *0.272* | *0.458* | *0.000* |
| | *bad # female* | *0.357* | *0.220* | *0.578* | *0.000* |
| **Kessler Psychological Distress Scale (K6)** | No or mild mental distress # male | ref. | | | |
| | *Moderate mental distress # male* | *0.777* | *0.659* | *0.917* | *0.003* |
| | *Serious mental distress # male* | *0.414* | *0.339* | *0.506* | *0.000* |
| | *No or mild mental distress # female* | *0.831* | *0.717* | *0.964* | *0.014* |
| | *Moderate mental distress # female* | *0.659* | *0.562* | *0.773* | *0.000* |
| | *Serious mental distress # female* | *0.393* | *0.326* | *0.474* | *0.000* |

community-based care in the recently planned community-based integrated long-term care indicated the importance of family caregiving. Policies are expected to enrich supports and benefits to family caregivers and to address the gender inequities in home-based caregiving [47, 48].

Besides physical and mental health status and home-based caregiving, we also confirmed the effects of having a conversation with others and the availability of social supports for various occasions on the positive longevity anticipation, as benefits of social relationship and social supports as well as negative effects of social isolation on population health and wellbeing were proved in previous studies at diverse settings [49–53]. Our results showed that different patterns of the association between social relationship and social supports in male and female: for a more positive attitude to a longer lifetime, male were more likely to be benefit from having reliable partner(s) for sharing joys and sorrows of life and for helps in daily life, while female were from having a conversation with others and having reliable partner(s) for nursing care. The finding further confirmed gender difference on family and social relations for life satisfaction [54, 55].

Regarding the financial status, our results indicated the significant effects of annual household income and household savings on the positive longevity anticipation, whereas findings beyond our hypotheses are that those with middle-level but not low-level income and those not qualified for the public assistance compared the counterparts, were less likely to have a positive attitude to the long-live. These findings seemed to be controversial to a previous finding that individuals with lower income status are more inclined to both perceived income inequality and less subjective wellbeing [56]. Considering the social context of Japan, the fact that those with low-level income and qualified for the public assistance are publicly subsidized for healthcare and long-term care and that the income level in the middle class is lowering and that inequalities are widening as the results of increasing temporary employment, job polarization and population aging may be explainable to these findings [57]. They reflected concerns on the late life arising from the expanding social inequality, though further researches are needed to examine the relevant mechanism.

In interpreting these major findings, several issues should be carefully considered. Above all, the cross-sectional study design limited the capacity to identify causalities and the longitudinal long-term effects. Moreover, the principal outcome was subjective and tentative. Because the questions were structured, qualitative approaches were of lack, in terms of exploring and interpreting the mechanism how the individual anticipation on longevity has emerged. Nevertheless, our study assessed the overall population of the country. As the national survey is

implemented in every five years, follow-up of this noteworthy issue and analysis based on panel data is expected to address these limitations.

## Conclusion

In conclusion, by using the latest dataset of the official national survey, this analysis adds knowledge on association of the individual desire regarding the length of lifespan and gender disparities in the growing concerns on the late life in the overall population of Japan. The findings are suggestive to reform the social security system in the super aged society, in particular in terms of promoting formal and informal caregiving functions, fostering social supports and dealing with inequalities.

## Supporting information

**S1 Checklist. The RECORD statement–checklist of items, extended from the STROBE statement, that should be reported in observational studies using routinely collected health data.**
(DOCX)

## Author Contributions

**Conceptualization:** Ruoyan Gai Tobe, Nobuyuki Izumida.

**Formal analysis:** Ruoyan Gai Tobe.

**Investigation:** Nobuyuki Izumida.

**Methodology:** Ruoyan Gai Tobe, Nobuyuki Izumida.

**Project administration:** Nobuyuki Izumida.

**Writing – original draft:** Ruoyan Gai Tobe.

**Writing – review & editing:** Ruoyan Gai Tobe, Nobuyuki Izumida.

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
