## [Decision Letter · Decision Letter 0]

9 Dec 2020

PONE-D-20-25313

Gender disparity in the individual attitude toward longevity among Japanese population: Findings from a national survey

PLOS ONE

Dear Dr. Gai,

Thank you for submitting your manuscript to PLOS ONE. After careful consideration, we feel that it has merit but does not fully meet PLOS ONE’s publication criteria as it currently stands. Therefore, we invite you to submit a revised version of the manuscript that addresses the points raised during the review process.

We look forward to receiving your revised manuscript.

Kind regards,

Thach Duc Tran, M.Sc., Ph.D.

Academic Editor

PLOS ONE

Journal Requirements:

2. As part of your revision, please complete and submit a copy of the RECORD checklist, a document that aims to improve reporting and reproducibility of observational studies that use routinely-collected data for purposes of post-publication data analysis and reproducibility: (http://record-statement.org). Please include your completed checklist as a Supporting Information file. Note that if your paper is accepted for publication, this checklist will be published as part of your article.

3. Please clarify whether the data utilized in this study were de-identified/anonymised before access?

4. Please revise your discussion and conclusions to emphasize on associations - and not causation/ determinant - between exposure variables and longevity desire. Due to susceptibility to bias and confounding and inability to determine temporal precedence, a cross-sectional study is not the appropriate study design for generating evidence for causation or determinants of outcomes.

5. Please include your tables as part of your main manuscript and remove the individual files. Please note that supplementary tables (should remain/ be uploaded) as separate "supporting information" files

6.Thank you for stating the following in the Acknowledgments Section of your manuscript:

[This study is granted by the Project of “The National Survey on Social Security and

People’s Life”.]

 [The author(s) received no specific funding for this work.]

7.We note that you have indicated that data from this study are available upon request. PLOS only allows data to be available upon request if there are legal or ethical restrictions on sharing data publicly. For information on unacceptable data access restrictions, please see http://journals.plos.org/plosone/s/data-availability#loc-unacceptable-data-access-restrictions.

Reviewers' comments:

Reviewer's Responses to Questions

**Comments to the Author**

1. Is the manuscript technically sound, and do the data support the conclusions?

Reviewer #1: Yes

2. Has the statistical analysis been performed appropriately and rigorously? 

Reviewer #1: Yes

3. Have the authors made all data underlying the findings in their manuscript fully available?

Reviewer #1: No

4. Is the manuscript presented in an intelligible fashion and written in standard English?

Reviewer #1: Yes

5. Review Comments to the Author

Reviewer #1: Societies are getting older around the world. Considering this fact, this study examines the anticipation to longevity from a gender perspective using data from Japan. This paper study a very interesting topic, nevertheless some changes may be made to improve it.

GENERAL

1. Including the number of pages and lines would help the revision process.

2. Please, check line by line. Some typos or problems in the word order can be found

INTRODUCTION

3. I would like to read something referring the characteristics of the Japanese society: the role of women, the role of families, the role older people play in society, the quality of support networks, the general feeling towards loneliness, the role of religion, legal framework towards gender equality etc. Have you think about including some of these?

METHODOLOGY

4. I would like to know if you have studied the impact of some variables such as:

a. The area: rural or urban. I suppose the prefectures and municipalities may have both areas, if not please correct me.

b. The access to services

c. The kind of home: flat, detached, semi-detached

d. Employment status

e. Do respondents receive help from the public institutions, cash transfers etc?

5. Why have you decided to dichotomize the answers into “strongly agree / somewhat agree versus strongly disagree / somewhat disagree”. Have you thought about the relationship between “somewhat agree” and “somewhat disagree”? The real feeling behind both answers can be determined by other factors while the “extreme” options can rely on stronger arguments

6. Have you tried another options instead of creating age groups? Have you tried to analyze the nonlinear relationship between age and the answers?

7. It could be interesting to include more participants’ information from the gender perspective to compare the characteristics of both groups.

RESULTS AND FINDINGS

8. Regarding the women/men comparison: Why do you report results on household income for men but the financial support for women? (This is just an example on two variables, but the comment could be extended to others) Maybe this can be explained by the role women play in Japan. Then, don’t you think women could not feel comfortable with this role and this can affect their feelings towards ageing?

9. The feeling towards ageing may vary with age. Are these variations the same for men and women? If they differ, how could you explain it?

10. Authors focus on the health status to explain the gender gap. I think that in this case, other factors are influencing respondents’ answers. Authors mention that “contextual influences” or “personal beliefs” are also relevant according to the literature. How do you think the women participation in the labor market could affect their attitude towards ageing?

6. PLOS authors have the option to publish the peer review history of their article (what does this mean?). If published, this will include your full peer review and any attached files.

Reviewer #1: No

---

## [Author Response · Author response to Decision Letter 0]

28 Jan 2021

Response to Journal Requirements

We highly appreciate these constructive comments from editors and reviewers and have revised the manuscript based on them with marks and responses colored orange.

→ The manuscript has been formatted following the suggested templates.

2. As part of your revision, please complete and submit a copy of the RECORD checklist, a document that aims to improve reporting and reproducibility of observational studies that use routinely-collected data for purposes of post-publication data analysis and reproducibility: (http://record-statement.org). Please include your completed checklist as a Supporting Information file. Note that if your paper is accepted for publication, this checklist will be published as part of your article.

→A RECORD checklist has been created as required.

3. Please clarify whether the data utilized in this study were de-identified/anonymised before access?

 →Yes the data utilized in this study were anonymized in the survey.

4. Please revise your discussion and conclusions to emphasize on associations - and not causation/ determinant - between exposure variables and longevity desire. Due to susceptibility to bias and confounding and inability to determine temporal precedence, a cross-sectional study is not the appropriate study design for generating evidence for causation or determinants of outcomes.

 →Related parts have been revised as suggested.

5. Please include your tables as part of your main manuscript and remove the individual files. Please note that supplementary tables (should remain/ be uploaded) as separate "supporting information" files

 →Tables were included in the main manuscript.

[This study is granted by the Project of “The National Survey on Social Security and

People’s Life”.]

 [The author(s) received no specific funding for this work.]

→ The statement on funding has been removed.

7. We note that you have indicated that data from this study are available upon request. PLOS only allows data to be available upon request if there are legal or ethical restrictions on sharing data publicly. For information on unacceptable data access restrictions, please see http://journals.plos.org/plosone/s/data-availability#loc-unacceptable-data-access-restrictions.

→ The data used in the study are available upon application to National Institute of Population and Social Security (IPSS), based on relevant regulations of Ministry of Health, Labor and Welfare of Japan on national statistics. Although the authors do not have the authorization on sharing data publicly, the data are available upon application for research purposes. For inquiry or request of the survey data, please contact the link below and mention “The National Survey on Social Security and People's Life 2017”.

http://www.ipss.go.jp/mail/e_sendmail/mail.html

This issue has been added in the cover letter as well.

 

Response to Reviewers' comments:

Reviewer's Responses to Questions

Comments to the Author

1. Is the manuscript technically sound, and do the data support the conclusions?

Reviewer #1: Yes

2. Has the statistical analysis been performed appropriately and rigorously? 

Reviewer #1: Yes

3. Have the authors made all data underlying the findings in their manuscript fully available?

Reviewer #1: No

4. Is the manuscript presented in an intelligible fashion and written in standard English?

Reviewer #1: Yes

5. Review Comments to the Author

Reviewer #1: Societies are getting older around the world. Considering this fact, this study examines the anticipation to longevity from a gender perspective using data from Japan. This paper study a very interesting topic, nevertheless some changes may be made to improve it.

GENERAL

1. Including the number of pages and lines would help the revision process.

→ The number of pages and lines have been added.

2. Please, check line by line. Some typos or problems in the word order can be found.

→ Spelling problems have been corrected.

INTRODUCTION

3. I would like to read something referring the characteristics of the Japanese society: the role of women, the role of families, the role older people play in society, the quality of support networks, the general feeling towards loneliness, the role of religion, legal framework towards gender equality etc. Have you think about including some of these?

→ As suggested, We have briefly added some of these background information in Introduction.

METHODOLOGY

4. I would like to know if you have studied the impact of some variables such as:

a. The area: rural or urban. I suppose the prefectures and municipalities may have both areas, if not please correct me.

→ In general, instead of rural / urban, municipalities are coded from 1 (metropolitan area) to 6 (remote area with the lowest level of population density) in official surveys. However, this information was deleted in the dataset used in the study for the ethical purpose.

b. The access to services

→ As the survey questionnaire did not include many parts of related information on the access to services, we were not able to analyze them.

c. The kind of home: flat, detached, semi-detached

→ We added variable on home ownership in the model.

d. Employment status

→ This variable has been added in the model.

e. Do respondents receive help from the public institutions, cash transfers etc?

→ The study was implemented by staffs of local health centers. To each respondent, a small gift instead of cash transfer was given.

5. Why have you decided to dichotomize the answers into “strongly agree / somewhat agree versus strongly disagree / somewhat disagree”. Have you thought about the relationship between “somewhat agree” and “somewhat disagree”? The real feeling behind both answers can be determined by other factors while the “extreme” options can rely on stronger arguments

→ The reason of dichotomizing was to avoid potential small size in disagree items to preform multilevel regression models, while as suggested we added a full description of the four items to capture diverse answers.

6. Have you tried another options instead of creating age groups? Have you tried to analyze the nonlinear relationship between age and the answers?

→ Yes, we had tried several options to create age groups and analyzed the nonlinear relationship between age and the target outcome and got the three-level age variable. It is able to explain the difference by age following the social context of Japan.

7. It could be interesting to include more participants’ information from the gender perspective to compare the characteristics of both groups.

→ The model included related variables in the questionnaire to capture the characteristics of both groups. As suggested above, in the revision, we have added employment status in the model to adjust the predict the overall proportion of the positive answer (it was not significant, though).

RESULTS AND FINDINGS

8. Regarding the women/men comparison: Why do you report results on household income for men but the financial support for women? (This is just an example on two variables, but the comment could be extended to others) Maybe this can be explained by the role women play in Japan. Then, don’t you think women could not feel comfortable with this role and this can affect their feelings towards ageing?

→ Based on the results as showed in Table 4, middle household income and no household savings in men are less likely to have positive attitude compared to alternatives but income and household savings in women did not have the significant impact. Rather, those not qualified to public assistance in women have higher proportion of the negative answer. It may reflect the fact that men’s attitude to long-live depends on earnings, while women’s attitude may be attributable to being financially secured, slightly different. To some extent it suggests the “ongoing” but not “completed” transition of productive and reproductive roles of men and women. Regarding women’s attitude, their proportion of the positive answer were lower compared to men.

9. The feeling towards ageing may vary with age. Are these variations the same for men and women? If they differ, how could you explain it?

→ The descriptive results indicated that the anticipation to longevity tended to decrease with age in general, while in multilevel analysis, age is significant in women, as those aged between 40-70 years are less likely to have the positive answer, but isn’t significant in men. Considering the social context in Japan, this may be explained by the fact that women in the age group 40-70 years play a major role in nursing care for the elderly, as discussed in our paper.

10. Authors focus on the health status to explain the gender gap. I think that in this case, other factors are influencing respondents’ answers. Authors mention that “contextual influences” or “personal beliefs” are also relevant according to the literature. How do you think the women participation in the labor market could affect their attitude towards ageing?

→ Based on the findings, we discussed the aspects of health, home-based caregiving roles, social support and financial status. We added employment status in the model as suggested (it was not significant, though).

 Regarding the impact of women’s participation in the labor market, though it’s not directly showed in our results due to complicated mechanisms, it potentially influences the social value on aging in various aspects. For example, at the macro level, it may improve women’s financial status, lead to a profound change in marriage and family pattern, in social network and in gender role on nursing care. Some of these may be positive while others may be negative to longevity.

 to publish the peer review history of their article (what does this mean?). If published, this will include your full peer review and any attached files.

Do you want your identity to be public for this peer review? For information about this choice, including consent withdrawal, please see our Privacy Policy.

Reviewer #1: No

---

## [Decision Letter · Decision Letter 1]

2 Mar 2021

PONE-D-20-25313R1

Gender disparity in the individual attitude toward longevity among Japanese population: Findings from a national survey

PLOS ONE

Dear Dr. Gai,

Thank you for submitting your manuscript to PLOS ONE. After careful consideration, we feel that it has merit but does not fully meet PLOS ONE’s publication criteria as it currently stands. Therefore, we invite you to submit a revised version of the manuscript that addresses the points raised during the review process.

We look forward to receiving your revised manuscript.

Kind regards,

Thach Duc Tran, M.Sc., Ph.D.

Academic Editor

PLOS ONE

Journal Requirements:

Reviewers' comments:

Reviewer's Responses to Questions

**Comments to the Author**

1. If the authors have adequately addressed your comments raised in a previous round of review and you feel that this manuscript is now acceptable for publication, you may indicate that here to bypass the “Comments to the Author” section, enter your conflict of interest statement in the “Confidential to Editor” section, and submit your "Accept" recommendation.

Reviewer #1: (No Response)

2. Is the manuscript technically sound, and do the data support the conclusions?

Reviewer #1: Yes

3. Has the statistical analysis been performed appropriately and rigorously? 

Reviewer #1: Yes

4. Have the authors made all data underlying the findings in their manuscript fully available?

Reviewer #1: No

5. Is the manuscript presented in an intelligible fashion and written in standard English?

Reviewer #1: Yes

6. Review Comments to the Author

Reviewer #1: The new version has included some improvements following the recommendations of reviewers. Nevertheless, there are still some comments to be made.

1. One of the comments regarding the dataset was the inclusion of a variable capturing the type of area (rural or urban). Authors argue the existence of a variable (coded 1-6) that captures this fact. Nevertheless, they have not included it due to “ethical concerns”. I do not understand the reason why including information about the level of population in an area would mean a problem with ethics.

2. What are the changes in the model and results after including some of the recommended variables?

3. Another important comment referred to the decision of dichotomize the answers into “strongly agree / somewhat agree versus strongly disagree / somewhat disagree”. Authors have included information without dichotomizing this variable. Authors argue the “potential small size” in some items. I cannot fully agree with this issue: 1. Strongly agree (N=4816), 2. Somewhat agree (N=8634), 3. Somewhat disagree (N=5287) and 4. Strongly Disagree (N=801). I still thing that dichotomizing maybe is not the best approach. Could you please explain it better, give more reasons for this or include some other studies that have performed the analysis in the same way?

4. I think that including changes in the manuscript including the information and the answers to the comments would be useful for future readers that may have similar doubts/concerns when reading you interesting paper.

7. PLOS authors have the option to publish the peer review history of their article (what does this mean?). If published, this will include your full peer review and any attached files.

Reviewer #1: No

---

## [Author Response · Author response to Decision Letter 1]

14 Mar 2021

Reviewer #1: The new version has included some improvements following the recommendations of reviewers. Nevertheless, there are still some comments to be made.

1. One of the comments regarding the dataset was the inclusion of a variable capturing the type of area (rural or urban). Authors argue the existence of a variable (coded 1-6) that captures this fact. Nevertheless, they have not included it due to “ethical concerns”. I do not understand the reason why including information about the level of population in an area would mean a problem with ethics.

→ An ethical concern refers to the fact that after cross calculations, the size in the category 6 region (the remote area with the lowest level of population density) may be too small to remain anonymization in reality.

2. What are the changes in the model and results after including some of the recommended variables?

→ As recommended, we added home ownership and employment status in the regression models and the results were not affected in the overall population and by gender (Table 3 and Table 4).

3. Another important comment referred to the decision of dichotomize the answers into “strongly agree / somewhat agree versus strongly disagree / somewhat disagree”. Authors have included information without dichotomizing this variable. Authors argue the “potential small size” in some items. I cannot fully agree with this issue: 1. Strongly agree (N=4816), 2. Somewhat agree (N=8634), 3. Somewhat disagree (N=5287) and 4. Strongly Disagree (N=801). I still thing that dichotomizing maybe is not the best approach. Could you please explain it better, give more reasons for this or include some other studies that have performed the analysis in the same way?

→ Apologies for deficient explanations. Regarding whether or not dichotomizing the answers, our concern is that statistical power would be weakened when cross calculating the answer of the longevity desire with a number of variables. For example, when cross-calculating the distribution of the answer in those qualified for the public assistance (a total of 237) by gender, some items end up with less than 5. Still, we realize the overview of the participants and the survey would be benefit from the advice, so we added the distribution of the answers without dichotomizing in Table 2.

4. I think that including changes in the manuscript including the information and the answers to the comments would be useful for future readers that may have similar doubts/concerns when reading you interesting paper.

→Based on the advice, we added “The trajectory of essential family functions in caregiving of the elderly people is being weakened”, as marked light green in Discussion. The major information from the Discussion were following the finds.

---

## [Decision Letter · Decision Letter 2]

14 Apr 2021

PONE-D-20-25313R2

Gender disparity in the individual attitude toward longevity among Japanese population: Findings from a national survey

PLOS ONE

Dear Dr. Gai,

Thank you for submitting your manuscript to PLOS ONE. After careful consideration, we feel that it has merit but does not fully meet PLOS ONE’s publication criteria as it currently stands. Therefore, we invite you to submit a revised version of the manuscript that addresses the points raised during the review process.

We look forward to receiving your revised manuscript.

Kind regards,

Thach Duc Tran, M.Sc., Ph.D.

Academic Editor

PLOS ONE

Reviewers' comments:

Reviewer's Responses to Questions

**Comments to the Author**

1. If the authors have adequately addressed your comments raised in a previous round of review and you feel that this manuscript is now acceptable for publication, you may indicate that here to bypass the “Comments to the Author” section, enter your conflict of interest statement in the “Confidential to Editor” section, and submit your "Accept" recommendation.

Reviewer #1: (No Response)

2. Is the manuscript technically sound, and do the data support the conclusions?

Reviewer #1: Partly

3. Has the statistical analysis been performed appropriately and rigorously? 

Reviewer #1: N/A

4. Have the authors made all data underlying the findings in their manuscript fully available?

Reviewer #1: No

5. Is the manuscript presented in an intelligible fashion and written in standard English?

Reviewer #1: Yes

6. Review Comments to the Author

Reviewer #1: I have to make some comments to this manuscript. As I said in previous stages, the subject the paper addresses is very interesting. Nevertheless, I feel that authors has not been able to address properly all my comments:

1. Regarding ethics: Ethics can be understood as an important concern when talking about individuals. I have several doubts about applying ethical concerns due to anonymization for regions.

2. Binary variables: It is true that problems can arise due a little N. I do not know whether authors have tried another approach for these variables. Moreover, I wonder why not including for example the variable rural vs urban using the coding 1-6 that is available by transforming these codes into 0-1?

3. Recommended variables: Authors state that they have included “home ownership” and “employment status” and final results were not affected. Nevertheless, authors have not included these regressions in the paper or an explanation of the reasons behind the fact described by them (new variables do not affect the dependent variable). The attitude towards of longevity is the same for a young person who has never worked than for a similar person who has always have a well-paid job?

7. PLOS authors have the option to publish the peer review history of their article (what does this mean?). If published, this will include your full peer review and any attached files.

Reviewer #1: No

---

## [Author Response · Author response to Decision Letter 2]

30 May 2021

1. Regarding ethics: Ethics can be understood as an important concern when talking about individuals. I have several doubts about applying ethical concerns due to anonymization for regions.

→ My apologies for bad clarifications, causing misunderstanding. The ethical concerns remain when the size in some cross calculations (in particular those living in the remote region) is relevantly small, potentially causing a risk to get the individual identified directly or indirectly by reference to identifiers such as the location and demographical / socioeconomic factors.

For example, when cross calculating the regions × basic factors and getting a relevantly small number in a cell, it turned out that, it would be possible to trace or predict a natural person who took part in the survey in the setting by referring these factors (age, annual income, house ownership, public assistance, etc), even though the individual had been anonymized. This issue is in particular taken prudently when handling this kind of government survey / statistics.

 For this reason, in the procedure for the application to the secondary survey data use, the regional variable were not included into the dataset for this study after all.

2. Binary variables: It is true that problems can arise due a little N. I do not know whether authors have tried another approach for these variables. Moreover, I wonder why not including for example the variable rural vs urban using the coding 1-6 that is available by transforming these codes into 0-1?

　→Regarding the dependent variable, as the reviewer suggested dichotomizing may not be the best approach, we once considered sticking with the original variable, while our concern remains in that statistical power would be weakened due to a little N. For this reason, it may be worthy to keep dichotomizing.

 As for variable “rural and urban”, I understand the reviewer recommended adding it, which is quite common in practice. However, I am afraid that this time we could not do so in the analysis, as the regional variable hadn’t been included into the dataset we used for this study when applying for the secondary analysis because of the ethical concern as mentioned above. My apologies for making confusions.

3. Recommended variables: Authors state that they have included “home ownership” and “employment status” and final results were not affected. Nevertheless, authors have not included these regressions in the paper or an explanation of the reasons behind the fact described by them (new variables do not affect the dependent variable). The attitude towards of longevity is the same for a young person who has never worked than for a similar person who has always have a well-paid job?

→As recommended, we further included home ownership and employment status to the regression model as the explanatory variables (as briefly described in Method) and their results were added in Table 3 (for the whole population) and Table 4 (for each male and female group). Although in the univariate analysis, the results suggested a significant association between the two new variables and the target variable, the multilevel analysis results indicated that these two variables were no longer significantly affecting the attitude to longevity, after adjusting confounding effects of those variables input in the model. Then based on the results, those showed significance (p<0.05) were considered to have an independent influence to the target variable, and consequently discussed and examined in Discussion.

Regarding whether “the attitude to longevity is the same for a young person who has never worked than for a similar person who has always have a well-paid job”, it is worthy note that our results need to be interpreted at the population level and in a cross-sectional perspective, but have limited power to predict the attitude at the individual level and in a longitudinal perspective. 

It may be rational to assume that a young person who has never worked is probably not having a positive desire compared to his/her counterpart with a well-paid job at the individual level, while looking at young persons as a group, those with middle-level income and those not qualified to the public assistance tended to be less likely to have a positive desire while employment status didn’t show a significant influence (it’s not to say the employment isn’t related to the attitude to longevity, but its effect is going to be diminished when adjusting confounding effects in the multilevel model). As mentioned in Discussion, these findings may be explainable by the fact those with low-level income and qualified for the public assistance are publicly subsidized for healthcare and long-term care, that the income level in the middle class is lowering, and that inequalities are widening as the results of increasing temporary employment, job polarization and population aging. It suggests concerns on the late life arising from the changing social context of Japan, though due to the cross-sectional design as mentioned in the limitations (the last paragraph of Discussion), these kinds of longitudinal effects cannot be simply captured by the current survey.

---

## [Decision Letter · Decision Letter 3]

5 Jul 2021

Gender disparity in the individual attitude toward longevity among Japanese population: Findings from a national survey

PONE-D-20-25313R3

Dear Dr. Gai,

We’re pleased to inform you that your manuscript has been judged scientifically suitable for publication and will be formally accepted for publication once it meets all outstanding technical requirements.

Kind regards,

Thach Duc Tran, M.Sc., Ph.D.

Academic Editor

PLOS ONE

Additional Editor Comments (optional):

Reviewers' comments:

Reviewer's Responses to Questions

**Comments to the Author**

1. If the authors have adequately addressed your comments raised in a previous round of review and you feel that this manuscript is now acceptable for publication, you may indicate that here to bypass the “Comments to the Author” section, enter your conflict of interest statement in the “Confidential to Editor” section, and submit your "Accept" recommendation.

Reviewer #1: All comments have been addressed

2. Is the manuscript technically sound, and do the data support the conclusions?

Reviewer #1: Yes

3. Has the statistical analysis been performed appropriately and rigorously? 

Reviewer #1: Yes

4. Have the authors made all data underlying the findings in their manuscript fully available?

Reviewer #1: No

5. Is the manuscript presented in an intelligible fashion and written in standard English?

Reviewer #1: Yes

6. Review Comments to the Author

Reviewer #1: (No Response)

7. PLOS authors have the option to publish the peer review history of their article (what does this mean?). If published, this will include your full peer review and any attached files.

Reviewer #1: No

---

## [Editor Report · Acceptance letter]

23 Jul 2021

PONE-D-20-25313R3 

Gender disparity in the individual attitude toward longevity among Japanese population: Findings from a national survey 

Dear Dr. Gai Tobe:

I'm pleased to inform you that your manuscript has been deemed suitable for publication in PLOS ONE. Congratulations! Your manuscript is now with our production department. 

Kind regards, 

on behalf of

Dr. Thach Duc Tran 

Academic Editor

PLOS ONE